# Effect of general anesthesia and controlled mechanical ventilation on pulmonary ventilation distribution assessed by electrical impedance tomography in healthy children

Milena S. Nascimento[1,2☯]*, Celso M. Rebello[1☯], Eduardo L. V. Costa[2,3☯], Leticia C. Corrêa[4☯], Glasiele C. Alcala[4☯], Felipe S. Rossi[1,4☯], Caio C. A. Morais[2☯], Eliana Laurenti[5‡], Mauro C. Camara[5‡], Marcelo Iasi[5‡], Maria L. P. Apezzato[5‡], Cristiane do Prado[1‡], Marcelo B. P. Amato[2‡]

1 Departamento Materno-Infantil, Hospital Israelita Albert Einstein, São Paulo, São Paulo, Brazil, 2 Divisão de Pneumologia, Departamento de Cardiologia–Instituto do Coração (INCOR) Hospital das Clínicas, HCFMUSP, Faculdade de Medicina, Universidade de São Paulo, São Paulo, SP, Brazil, 3 Instituto de Ensino e Pesquisa—Hospital Sírio Libanês, São Paulo, São Paulo, Brazil, 4 Developer Division, Timpel SA, São Paulo, São Paulo, Brazil, 5 Departamento Centro Cirúrgico, Hospital Israelita Albert Einstein, São Paulo, São Paulo, Brazil

☯ These authors contributed equally to this work.
‡ EL, MCC, MI, MLPA, CP and MBPA also contributed equally to this work.
* milenasn@einstein.br

**Data Availability Statement:** All relevant data are within the manuscript and its Supporting Information files.

## Abstract

### Introduction

General anesthesia is associated with the development of atelectasis, which may affect lung ventilation. Electrical impedance tomography (EIT) is a noninvasive imaging tool that allows monitoring in real time the topographical changes in aeration and ventilation.

### Objective

To evaluate the pattern of distribution of pulmonary ventilation through EIT before and after anesthesia induction in pediatric patients without lung disease undergoing nonthoracic surgery.

### Methods

This was a prospective observational study including healthy children younger than 5 years who underwent nonthoracic surgery. Monitoring was performed continuously before and throughout the surgical period. Data analysis was divided into 5 periods: induction (spontaneous breathing, SB), ventilation-5min, ventilation-30min, ventilation-late and recovery-SB. In addition to demographic data, mechanical ventilation parameters were also collected. Ventilation impedance (Delta Z) and pulmonary ventilation distribution were analyzed cycle by cycle at the 5 periods.

### Results

Twenty patients were included, and redistribution of ventilation from the posterior to the anterior region was observed with the beginning of mechanical ventilation: on average, the

**Funding:** The authors received no specific funding for this work.

**Competing interests:** Letícia C. Corrêa and Glasiele C. Alcala are employees of Timpel S.A.; Eduardo Leite and Felipe S. Rossi are Timpel S.A. consultants, Marcelo B. P. Amato is Timpel S.A. consultant and minority shareholder. The other authors declare no competing interests. This does not alter our adherence to PLOS ONE policies on sharing data and materials.

percentage ventilation distribution in the dorsal region decreased from 54%(IC95%:49–60%) to 49%(IC95%:44–54%). With the restoration of spontaneous breathing, ventilation in the posterior region was restored.

## Conclusion

There were significant pulmonary changes observed during anesthesia and controlled mechanical ventilation in children younger than 5 years, mirroring the findings previously described adults. Monitoring these changes may contribute to guiding the individualized settings of the mechanical ventilator with the goal to prevent postoperative complications.

## Introduction

General anesthesia has been shown to affect respiratory function in adults, leading to atelectasis and redistribution of ventilation [1, 2]. These changes persist into the postoperative period and are associated with an increased frequency of postoperative respiratory complications [3].

Decreased muscle tone, especially of the diaphragm, is one of the main causes of the redistribution of pulmonary ventilation and loss of functional residual capacity (FRC) during controlled mechanical ventilation (CMV). The decreased tone causes a cephalad shift of the diaphragm, predominantly in its posterior region, compressing both lung and airways [4, 5]. This external compression together with a tendency to collapse of small airways at lower lung volumes is responsible for the decreased ventilation in the posterior regions and the ensuing atelectasis during CMV [5, 6].

Although it is well known that FRC decreases during the induction of anesthesia, the degree of this reduction and its determinants are less studied, probably due to difficulties related to estimating FRC and atelectasis at the bedside [6]. The paucity of data is especially true in the pediatric population. Electrical impedance tomography (EIT) imaging may play an important role in this context because it is portable, making its use feasible in the operating room (OR) [7, 8]. In adults, EIT has been used in the OR to monitor pulmonary ventilation and even to guide intraoperative ventilatory settings [9, 10] in a strategy that helped improve intraoperative oxygenation and prevent postoperative atelectasis [10, 11]. The use of EIT in pediatrics as a method of evaluating ventilation has gained ground in recent years, and the focus of most studies has been on the monitoring of regional changes in pulmonary aeration, with the goal of to improve knowledge about pulmonary physiology [12, 13] and to evaluate the impact of therapeutic interventions [14, 15].

Given the scarcity of studies in the pediatric population evaluating the effect of anesthesia and mechanical ventilation on the distribution of pulmonary ventilation in children with normal lungs, we decided to conduct a study to describe the pattern of ventilation distribution in pediatric patients without lung disease.

## Objectives

The aim of this study was to evaluate the changes in the distribution of pulmonary ventilation in pediatric patients without lung disease undergoing nonthoracic surgery. Our hypothesis is that anesthesia will produce dorsal atelectasis, which will lead to redistribution of ventilation from the posterior to the anterior region.

## Materials and methods

### Type and location of the study

A prospective observational clinical study was conducted in the OR of quaternary hospital, from August 2018 to December 2019, including children younger than 5 years without pulmonary disease undergoing nonthoracic surgery who required mechanical ventilation during surgery.

### Ethical aspects

This study was submitted, reviewed and either approved to the Ethics and Research Committee of Israelita Albert Einstein Hospital and to the Ethics and Research Committee of the School of Medicine of the University of São Paulo. An informed consent form was obtained from each child's parents.

The Ethics and Research Committee of Israelita Albert Einstein Hospital, approval number: 89911218.6.0000.0071, approval date: July 05, 2018.

The Ethics and Research Committee of the School of Medicine of the University of São Paulo, approval number: 89911218.6.3001.0065, approval date: November 28, 2019.

This study is in accordance with the recently amended Declaration of Helsinki of 1975. The risk of participating in this study was considered minimal, limited to the accidental loss of confidentiality of the collected data or to erythema on the skin at the site of application of the EIT belt. Patients were only monitored with EIT after their next of kin signed the informed consent form.

This study had partial results previously published in *Anesthesiology* [16].

### Inclusion and exclusion criteria

Children younger than 5 years without lung disease who underwent nonthoracic surgery were eligible for the study. Exclusion criteria included patients whose parents did not agree with their participation in the study. Patients who had serious and unexpected complications during the anesthetic induction process would be withdrawn from the study.

### Electrical impedance tomography

Data acquisition was performed using the Electrical Impedance Tomography—Plethysmography and Electrical Impedance Lung Function Analyzer (Enlight 1800—Timpel®, São Paulo, Brazil). In addition, a proximal pneumotachograph was used that fed a ventilation board coupled to the EIT, providing flow, pressure and tidal volume information.

For data collection, belts with 32 electrodes (for thoracic perimeters from 37.5 to 49.9 cm) or with 24 electrodes (for thoracic perimeters from 50 to 65.9 cm) were positioned around the thorax at the level of the intermammillary line. Through these electrodes, a low electrical current (5–10 mA) was applied, and the resistivity of the tissue to the passage of this current was estimated by an imaging reconstruction algorithm. The amount of air within a region has a linear relationship with its resistivity. The real-time images of resistivity can thus be used to monitor pulmonary aeration according to areas of interest of the lung.

The variables measured from the EIT acquisitions were tidal variations in lung impedance (Delta Z, a surrogate for regional tidal volume) and regional and global ventilation analyses were performed. As usual for this technique, EIT was measured in arbitrary units (AU) for delta Z and percentage for pulmonary ventilation distribution.

For regional analysis, the image was divided into two equal regions of interest (ROIs), anterior and posterior.

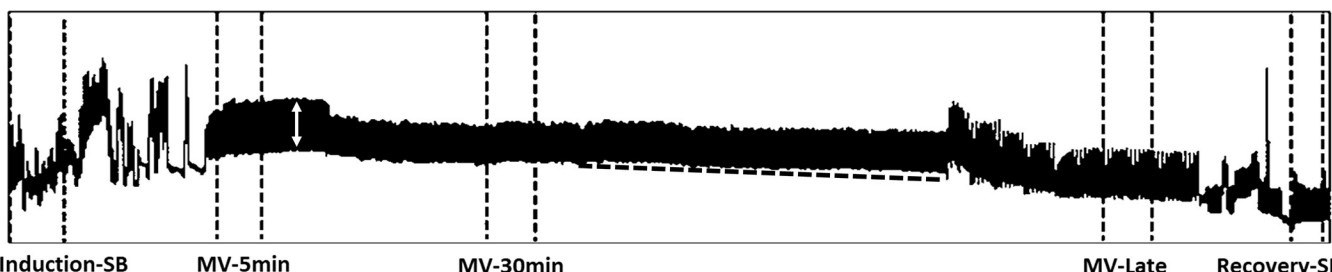

**Fig 1. Plethysmogram representing the total monitoring time with EIT.** The white tracing represents the variation in ventilation over time. Delta Z (double arrow) represents the tidal variation in impedance and is associated with tidal volume. The vertical dashed lines indicate the periods studied during monitoring: induction-SB (spontaneous breathing before anesthesia induction), MV-5min (initial five minutes of controlled ventilation), MV-30min (after 30 minutes of controlled ventilation), MV-late (final 5 minutes of controlled ventilation) and recovery-SB (after removal of the artificial airway and resumption of spontaneous breathing).

## Protocol

Continuous EIT monitoring started before anesthesia induction, maintained throughout the intraoperative period and was only interrupted a few minutes after the resumption of spontaneous breathing. For data analysis, the surgical time was divided into five periods: induction-SB (spontaneous breathing, SB), ventilation-5min (MV-5min, initial five minutes of controlled ventilation), ventilation-30min (MV-30min), ventilation-late (MV-late) final five minutes of controlled ventilation), and recovery-SB (after removal of the artificial airway and resumption of spontaneous breathing) (Fig 1).

## Analysis of physiological parameters and ventilatory assistance

Vital signs, including respiratory rate (RR), heart rate (HR), blood pressure (BP) and pulse oxygen saturation (SpO$_2$) were obtained using the interconnect monitor MP70 (Philips Medical Systems, The Netherlands). Data were collected on the type of ventilatory assistance used (spontaneous breathing or mechanical ventilation), interface used during mechanical ventilation (laryngeal mask or orotracheal tube) and mechanical ventilation parameters. As this was an observational study, there was no protocol for adjusting ventilation parameters. They were adjusted according to the anesthesiologist.

During ventilation (MV-5min, MV-30min and MV-late), the parameters collected were inspiratory pressure (P$_{insp}$), positive end-expiratory pressure (PEEP), the mean airway pressure (MAP), respiratory rate (RR), inspiratory time (t$_{insp}$), fraction of inspired oxygen (FiO$_2$), and absolute tidal volume (V$_T$). The delta pressure (P$_{ins}$—PEEP), V$_T$ adjusted for patient weight (in mL/kg) and compliance (mL/cmH$_2$O) were calculated.

## Ventilation impedance and distribution of pulmonary ventilation—EIT

The analysis was performed off-line using a proprietary software developed by Timpel SA (Offline Analysis, 2019). A period of five consecutive minutes was used for the analysis of the different periods. For the induction-SB and recovery-SB periods, the highest number of breaths of each moment was used without the use of positive pressure.

Calculations to obtain Delta Z were performed by subtracting the inspiration value from the expiration value. The distribution of ventilation in percentage was presented as dorsal ventilation fraction, which was calculated by the ratio of the Delta Z of the posterior region by the global Delta Z multiplied by 100 and by center of ventilation (CoV) that is defined based on the center of gravity or center of mass and its value is given automatically by the EIT [17].

The measurements of Delta Z (ventilation) in the anterior/posterior regions were also compared in the five periods.

## Statistical analysis

The data were described as absolute and relative frequencies for the categorical variables and as medians and quartiles, in addition to minimum and maximum values for the numerical variables.

The means of the mechanical ventilation parameters were obtained by fitting models of generalized estimation equations. Changes in global, regional ventilation and CoV were tested with linear mixed models, accounting for the repeated measurements with a random intercept. In addition, models were adjusted with subjects nested within time (induction-SB, MV-5min, MV-30min, MV-late, recovery-SB), considering the multiple measurements (breaths) from the same subject at each time point. The results of the models are presented as the mean values estimated by the models with corresponding 95% confidence intervals and $p$ values [18]. The analyses were performed with the R statistical software (R Foundation for Statistical Computing, Vienna, Austria) considering a significance level of 5%. As there is no reference to pediatric values of the EIT data evaluated for sample calculation, it was decided to carry out a convenience sample.

## Results

A total of 21 pediatric patients undergoing nonthoracic surgery with general anesthesia were included. One patient was excluded due to an error in the reading of the software, leaving 20 patients for analysis. No patient was withdrawn due to complications in the surgical period. The demographic data of 20 patients, as well as the type of surgery and type of interface used, are shown in Table 1.

Anesthetic induction was performed through inhalation anesthesia with sevoflurane in 19 patients (95%), and this was maintained throughout the surgery. Propofol (100%), fentanyl

**Table 1. Demographic characteristics of pediatric patients evaluated using EIT (n = 20).**

|  | n (%) | Median (Q1; Q3) | Min; Max |
|---|---|---|---|
| **Gender** |  |  |  |
| **Male** | 18 (90%) |  |  |
| **Age (months)** |  | 27.5 (13.9; 49.4) | 1.9; 61.5 |
| **Weight (kg)** |  | 13.0 (9.9; 16.5) | 5.3; 22.0 |
| **Height (cm)** |  | 87.0 (75.0; 101.0) | 53.0; 115.0 |
| **Perimeter of the thorax (cm)** |  | 53.5 (48.8; 55.5) | 39.0; 60.0 |
| **Type of surgery** |  |  |  |
| Postectomy | 16 (80%) |  |  |
| Unilateral inguinal hernioplasty | 4 (20%) |  |  |
| Bilateral inguinal hernioplasty | 2 (10%) |  |  |
| Umbilical hernioplasty | 1 (5%) |  |  |
| Epigastric hernioplasty | 1 (5%) |  |  |
| Cholecystectomy | 1 (5%) |  |  |
| **Ventilatory interface** |  |  |  |
| Laryngeal mask airway | 13 (65%) |  |  |
| OT | 7 (35%) |  |  |

Q1: first quartile; Q3: third quartile OT: orotracheal tube

**Table 2. Mechanical ventilation parameters in 20 pediatric patients, by time of collection.**

| Parameters | Periods | | | p |
|---|---|---|---|---|
| | MV-5min | MV-30min | MV-late | |
| $P_{insp}$ (cmH$_2$O) | 15 (13; 17) | 15 (13; 17) | 13 (10; 17) | 0.576 |
| PEEP (cmH$_2$O) | 4 (3; 4) | 4 (3; 5) | 3 (2; 5) | 0.662 |
| Delta P (cmH$_2$O) | 11 (10; 13) | 11 (10; 13) | 10 (7; 13) | 0.592 |
| MAP (cmH$_2$O) | 8 (7; 9) | 8 (6; 9) | 7 (5; 9) | 0.655 |
| RR (per min) | 22 (20; 25) | 22 (20; 25) | 22 (20; 25) | 0.848 |
| $T_{insp}$ (s) | 1.02 (0.92; 1.13) | 0.97 (0.89; 1.06) | 1.01 (0.85; 1.19) | 0.566 |
| FiO$_2$ (%) | 48 (42; 54) | 47 (43; 51) | 48 (44; 53) | 0.193 |
| $V_T$ (mL) | 143 (121; 169) | 152 (121; 190) | 151 (126; 182) | 0.787 |
| $V_T$ (mL/kg) | 10 (9; 12) | 10 (9; 11) | 11 (10; 11) | 0.852 |
| C (ml/cmH$_2$O) | 13,58 (11,18; 16,51) | 13,54 (11,20; 16,37) | 12,97 (10,63; 15,84) | 0.680 |

$P_{insp}$ = inspiratory pressure, PEEP = positive end-expiratory pressure, Delta P = delta pressure, MAP = mean airway pressure, RR = respiratory rate, $t_{insp}$ = inspiratory time, FiO$_2$ = fraction of inspired oxygen, $V_T$ = absolute tidal volume, $V_T$ tidal volume adjusted for patient weight, C = compliance

(95%) and atropine (15%) were used for rapid sequence intubation. Neuromuscular blockers were used in only 2 patients: one received succinylcholine, and the other received rocuronium. Epidural block was performed in 3 (15%) patients with ropivacaine and in 15 (75%) patients who underwent penile block, 2 with bupivacaine and 13 with levobupivacaine. Some patients were submitted to more than one surgical procedure during anesthesia.

Pressure Controlled ventilatory modality was used in all patients and no patient triggered cycle was detected during the analyzed periods. The ventilation parameters during the 3 periods (MV-5min, MV-30min and MV-late) are shown in Table 2. There was no difference in the parameters during the three periods of mechanical ventilation.

The total ventilation time was greater than 30 minutes for eight patients who were monitored at the MV-late time and had a median of 62.5 minutes (IC%: 53.75; 76.25). For the others that had a total ventilation time of 30 minutes, the MV-30min and MV-late periods are coincident, and therefore, only the measurements in MV-30 were considered.

There was a redistribution of ventilation from the posterior to the anterior region with the onset of MV (posterior ventilation went from 54% (IC95: 49–60%) to 49% (IC95:44–53%) (p = 0.002) and remained stable over the whole ventilation period (Fig 2). With the restoration of spontaneous breathing, ventilation distribution returned towards the normal pattern with predominant ventilation in the dorsal regions (posterior ventilation went from 51% (IC95: 42–61%) to 56% (IC95:51–62%), although a small difference persisted as compared to the induction-SB period but without significance (Fig 2).

When ventilation distribution was evaluated by CoV the results were like those found by the dorsal ventilation fraction. There was redistribution from the posterior to the anterior region observed with the onset of the MV (posterior ventilation went from 51% (IC95: 49–53%) to 47% (IC95:46–49%) (p<0.001). With the restoration of spontaneous breathing, ventilation distribution returned towards the normal pattern with predominant ventilation in the dorsal regions (posterior ventilation went from 48% (IC95: 47–50%) to 49% (IC95:47–51%), although a small difference persisted as compared to the induction-SB period but without significance (Fig 3).

For the measures of impedance variation, represented by the Delta Z in arbitrary units, significant variations were observed in the measurements of global and anterior Delta Z compared all mechanical ventilation time to induction-SB (p <0.01). After the beginning of the positive pressure of mechanical ventilation (mean of Pinsp = 15 cmH$_2$O), there was an increase

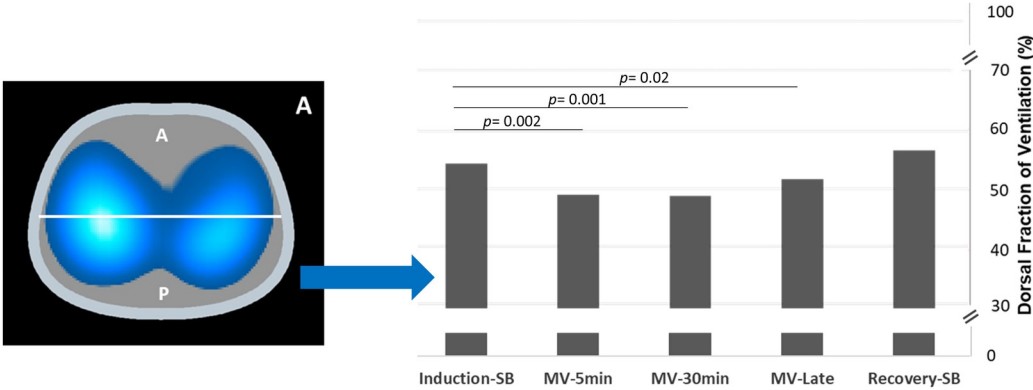

**Fig 2. Distribution of pulmonary ventilation assessed by the dorsal ventilation fraction in the anterior and posterior regions presented over the five time points: Induction-SB, MV-5min, MV-30min, MV-late and recovery-SB.** The dorsal ventilation fraction is calculated by the ratio of the Delta Z of the posterior region to the global Delta Z multiplied by 100. Significant variations were observed in the ventilation distribution measures demonstrated in percentage when compared MV-5min, MV-30min and MV-late to induction-SB.

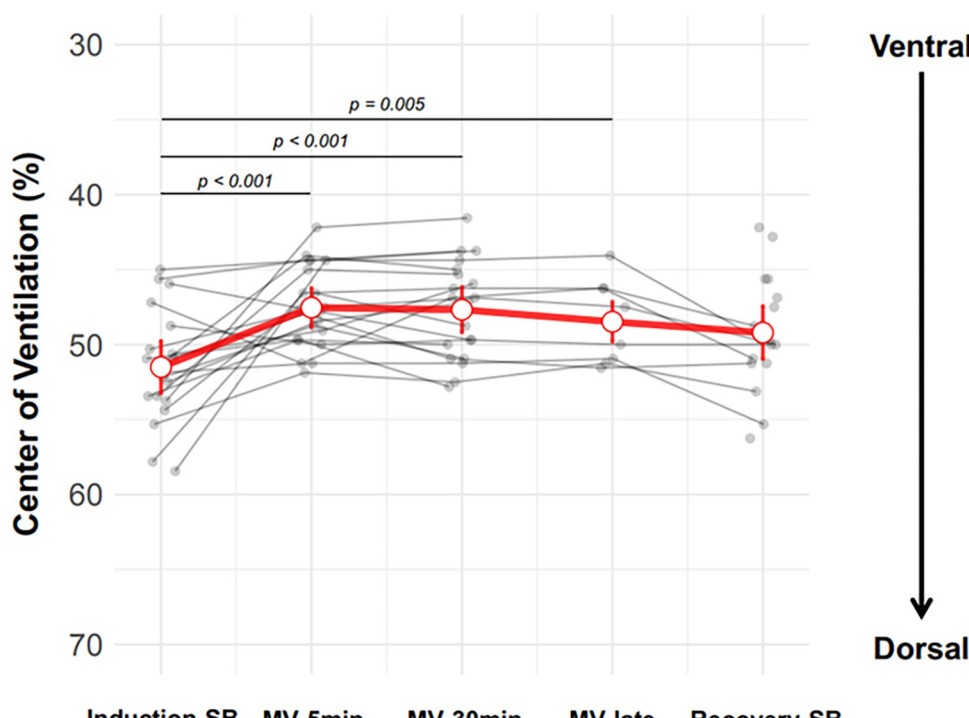

**Fig 3. Distribution of pulmonary ventilation assessed by center of ventilation (CoV) presented over the five time points: Induction-SB, MV-5min, MV-30min, MV-late and recovery-SB.** The CoV is a parameter that quantifies the distribution of ventilation. When we have a value of 50 it means that ventilation is equally distributed between the anterior and posterior regions of the thorax. Higher numbers indicate a shift towards the dorsal region, and lower numbers indicate a shift towards the ventral region. Significant variations were observed in the ventilation distribution measures demonstrated in percentage when compared MV-5min, MV-30min and MV-late to induction-SB.

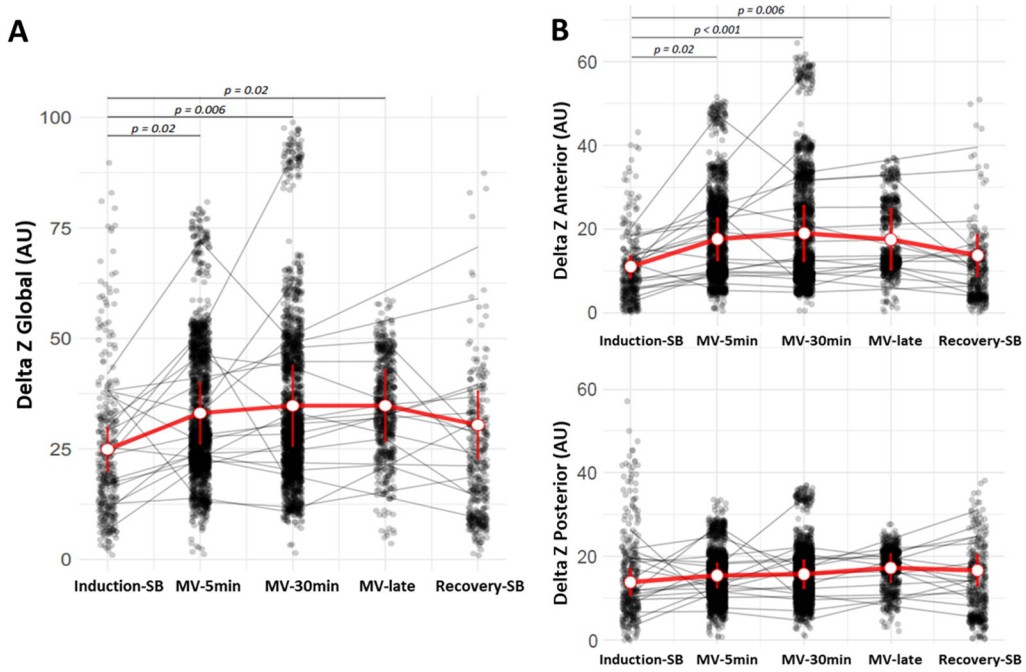

**Fig 4.** Impedance change (Delta Z) in the global (A) and in the anterior (B) and posterior (C) regions over the five periods: induction-SB, MV-5min, MV-30min, MV-late and recovery-SB. Significant variations were observed in the measurements of global and anterior Delta Z compared all mechanical ventilation time to induction-SB. Gray dots represent multiple measurements (respiratory cycles) at each time point from all patients. Gray lines describe the average behavior from each patient. Error bars represent the confidence interval with mean (open red circle).

in Delta Z, which maintained during all mechanical ventilation. After the removal of positive pressure and the resumption of spontaneous breathing, a drop in the Delta Z was observed. Significant variations were not observed in the measurement posterior Delta Z (Fig 4).

## Discussion

Our main findings were that, in lung-healthy children, general anesthesia together with mechanical ventilation led to redistribution of pulmonary ventilation from the posterior to the anterior region when compared to spontaneous breathing in the preoperative period.

We showed that the distribution of ventilation at the time of anesthesia induction, when patients were breathing spontaneously, was greater in the posterior region, where approximately 60% of ventilation concentrated. After the beginning of CMV, there was redistribution of ventilation to the anterior region, which then received half the ventilation. This pattern has been described in adults in whom the diaphragm relaxation appears to be the main physiological mechanism responsible for this change [5, 6]. During spontaneous breathing, diaphragmatic contraction has a strong influence on the direction of ventilation. The posterior portion of the dome presents greater amplitude during the incursion and, therefore, greater pressure variation in the posterior region of the thorax. During active inspiration, the dorsal part of the diaphragm has a greater range of incursion. In this region, the muscle fibers are more elongated, therefore, they can develop more strength during inspiration. With greater amplitude of incursion, the variation of pleural pressure is greater, leading to greater ventilation of the posterior region. However, as soon as the muscle relax take, ventilation is shifted toward anterior region. That because is during muscle relax and in supine position, the anterior chest wall is more readily expanded as the posterior chest wall is splinted by the surface the patient is lying

on and the cranial movement of the anterior diaphragm. During anesthesia of adult patients, the shape of the chest changes with a reduction in the cross-sectional area of the ribcage. The decrease in the tone of the diaphragm during CMV causes its cephalad movement especially of the most dorsal parts. There is loss of ventilation in these posterior regions with a consequent increase in the ventilation of the anterior regions [5, 6, 19]. The use of spontaneous breathing during mechanical ventilation has been discussed as a strategy to prevent the redistribution of ventilation in dependent zones, however, with controversial results [20, 21].

In the pediatric population, the distribution of ventilation in spontaneous breathing has great variability because it is influenced not only by the variation in pleural pressure but also by the instability of small airways, factors that influence the distribution of ventilation between dependent and nondependent regions [14, 22]. There was doubt as to whether the redistribution of ventilation during mechanical ventilation observed in adults would also occur in the pediatric age group. Our study evaluated the transition from spontaneous breathing to CMV and we confirmed that a shift to the anterior regions occurred, like that described in adults.

Many studies in children involving mechanical ventilation and EIT included patients with lung diseases [23–25] The evaluation of the impact of mechanical ventilation on changes in ventilation distribution is influenced by ventilation itself and by the underlying disease. In our patients, we observed the isolated effect of CMV, since only patients with healthy lungs were included.

In our study, we present the distribution of ventilation as a dorsal ventilation fraction [26, 27], as described by Yoshida T et al., and by center of ventilation (CoV) as described by Frerichs et al [17]. The CoV is a position within the image plane and can be presented as Cartesian coordinates or as percentages, with 0 representing one extreme of the image and 100% the other extreme, in both axes. Conceptually, the CoV is the point or fulcrum that balances all ventilation in the image plane. Despite the differences in implementation, the two measures, i.e., the dorsal ventilation fraction and the CoV, capture the same phenomenon [17]. Our group believes that the concept of dorsal ventilation fraction is more intuitive than the CoV, but it is important to have both concepts in mind, as different publications might use one or the other to assess the ventilation directed to the dependent and the non-dependent lung areas. Our study, in addition to evaluating the transition from spontaneous breathing to mechanical ventilation, also monitored the entire surgical time until the restoration of spontaneous breathing in the recovery period. This made it possible to evaluate the impact of the time of CMV with inactivity of the diaphragm on the redistribution of ventilation.

The analysis of the impedance variation, Delta Z, showed an increase in whole ventilation with the beginning of mechanical ventilation, a fact previously described once the use of positive inspiratory pressure results in higher tidal volume compared to spontaneous breathing [20–28]. Humphreys et al evaluating children undergoing cardiac surgery showed that the Delta Z almost doubles with the start of mechanical ventilation when compared to spontaneous breathing [28]. The increase in Delta Z in the nondependent region should serve as a warning because this region will be prone to ventilation-induced lung injury during anesthesia. We emphasize that the use of EIT during anesthesia and mechanical ventilation can contribute to individualized ventilatory assistance and assist in the prevention of complications due to poorly adjusted parameters. Once we know that in children with healthy lungs the anterior-posterior ventilation distribution during mechanical ventilation is quite similar, we might titrate the ventilatory parameters to maintain this distribution pattern. In accordance with this concept, if posterior zones ventilation is going below 50%, collapse might be present and then PEEP increase and FRC gain shall be the target. On the other hand, if posterior ventilation distribution is above 50%, anterior hyperinflation is expected and parameters adjustments such as PEEP reduction or tidal volume adjustment are likely necessary.

Our study has several limitations. First, it was an observational, descriptive study, and as such we cannot draw any conclusions about the influence on the variable studied of the ventilatory parameters used during the surgical procedure. Second, this was a single center study, and finally, sample size was relatively small.

## Conclusion

We demonstrated that ventilation shifts ventrally during controlled mechanical ventilation. The pulmonary changes observed during anesthesia and controlled mechanical ventilation in children follow the same repercussions found in adults. These findings may contribute to guiding the individualized configurations of the ventilator and even preventing postoperative complications.

## Supporting information

**S1 Data.**
(XLSX)

## Acknowledgments

We thank the statistics team, especially for the help of Sandra Regina Malagutti and the entire team at the surgical center.

## Author Contributions

**Conceptualization:** Milena S. Nascimento, Celso M. Rebello, Eduardo L. V. Costa, Leticia C. Corrêa, Glasiele C. Alcala, Felipe S. Rossi, Caio C. A. Morais, Marcelo B. P. Amato.

**Formal analysis:** Milena S. Nascimento, Celso M. Rebello, Eduardo L. V. Costa, Leticia C. Corrêa, Glasiele C. Alcala, Caio C. A. Morais, Cristiane do Prado, Marcelo B. P. Amato.

**Investigation:** Eliana Laurenti, Mauro C. Camara, Marcelo Iasi, Maria L. P. Apezzato.

**Methodology:** Milena S. Nascimento, Celso M. Rebello, Eduardo L. V. Costa, Leticia C. Corrêa, Glasiele C. Alcala, Felipe S. Rossi, Caio C. A. Morais, Eliana Laurenti, Mauro C. Camara, Marcelo Iasi, Maria L. P. Apezzato, Cristiane do Prado, Marcelo B. P. Amato.

**Validation:** Milena S. Nascimento, Celso M. Rebello, Eduardo L. V. Costa, Felipe S. Rossi, Caio C. A. Morais, Eliana Laurenti, Mauro C. Camara, Marcelo Iasi, Maria L. P. Apezzato, Cristiane do Prado, Marcelo B. P. Amato.

**Writing – original draft:** Milena S. Nascimento, Celso M. Rebello, Eduardo L. V. Costa, Leticia C. Corrêa, Glasiele C. Alcala, Felipe S. Rossi, Eliana Laurenti, Mauro C. Camara, Marcelo Iasi, Maria L. P. Apezzato, Cristiane do Prado, Marcelo B. P. Amato.

**Writing – review & editing:** Milena S. Nascimento, Celso M. Rebello, Eduardo L. V. Costa, Leticia C. Corrêa, Glasiele C. Alcala, Felipe S. Rossi, Caio C. A. Morais, Eliana Laurenti, Mauro C. Camara, Marcelo Iasi, Maria L. P. Apezzato, Cristiane do Prado, Marcelo B. P. Amato.

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
