## [Decision Letter · Decision Letter 0]

1 Sep 2022

PONE-D-22-15065Effect of anesthesia and controlled mechanical ventilation on pulmonary ventilation distribution assessed by EIT in healthy childrenPLOS ONE

Dear Dr. Nascimento,

Thank you for submitting your manuscript to PLOS ONE. After careful consideration, we feel that it has merit but does not fully meet PLOS ONE’s publication criteria as it currently stands. Therefore, we invite you to submit a revised version of the manuscript that addresses the points raised during the review process.

Please revise. 

We look forward to receiving your revised manuscript.

Kind regards,

Academic Editor

PLOS ONE

Journal Requirements:

2. You indicated that you had ethical approval for your study. In your Methods section, please ensure you have also stated whether you obtained consent from parents or guardians of the minors included in the study or whether the research ethics committee or IRB specifically waived the need for their consent

Reviewers' comments:

Reviewer's Responses to Questions

**Comments to the Author**

1. Is the manuscript technically sound, and do the data support the conclusions?

Reviewer #1: Yes

2. Has the statistical analysis been performed appropriately and rigorously? 

Reviewer #1: Yes

3. Have the authors made all data underlying the findings in their manuscript fully available?

Reviewer #1: No

4. Is the manuscript presented in an intelligible fashion and written in standard English?

Reviewer #1: Yes

5. Review Comments to the Author

Reviewer #1: The manuscript is well written and easy to understand. The study was an observational clinical study with simple and clear objectives. The authors report their findings about the development of atelectasis in the dorsal area in mechanically-ventilated patients with normal lungs who underwent nonthoracic surgery. They found that ventilation shifted from posterior to anterior region in the patients younger than 5 years as often observed in adult patients.

(1) The statistical analysis section is not clear. Explain the methods more clearly with citations. Rationales for the chosen statistical methods would be helpful. For example, it is difficult to understand the following expressions:

- “models of generalized estimation equations with gamma or Poisson distribution and logarithmic link function”

- “mixed linear models with identity or logarithmic link functions according to the best fit, considering the dependence between the various measurements performed at each evaluation time and between the moments evaluated in the same patient”

(2) Please provide explanations for greater ventilation in the posterior region during SB before anesthesia.

(3) Please provide more explanations about the increased Delta Z during CMV. What was the mode of MV?

(4) Please provide more explanations about the still-increased Delta Z during SB after surgery.

6. PLOS authors have the option to publish the peer review history of their article (what does this mean?). If published, this will include your full peer review and any attached files.

Reviewer #1: No

---

## [Author Response · Author response to Decision Letter 0]

28 Oct 2022

To the editor:

Dear editor, 

We would like to request the inclusion of an colaborator who participated in all stages of the study and unfortunately, we forgot to include at the time of submission: Caio C.A. Morais (Affiliation: Divisão de Pneumologia, Departamento de Cardiologia – Instituto do Coração (INCOR) Hospital das Clínicas, HCFMUSP, Faculdade de Medicina, Universidade de São Paulo, São Paulo, SP, BR).

Reviewers' Comments to Authors:

Reviewer #1: The manuscript is well written and easy to understand. The study was an observational clinical study with simple and clear objectives. The authors report their findings about the development of atelectasis in the dorsal area in mechanically ventilated patients with normal lungs who underwent nonthoracic surgery. They found that ventilation shifted from posterior to anterior region in the patients younger than 5 years as often observed in adult patients.

We appreciate the reviewers' time and effort dedicated to our manuscript, their comments and suggestions have certainly helped us improve our work. Below is a point-by-point answer to all the questions raised by the reviewers.

(1) The statistical analysis section is not clear. Explain the methods more clearly with citations. Rationales for the chosen statistical methods would be helpful. For example, it is difficult to understand the following expressions:

- “models of generalized estimation equations with gamma or Poisson distribution and logarithmic link function”

- “mixed linear models with identity or logarithmic link functions according to the best fit, considering the dependence between the various measurements performed at each evaluation time and between the moments evaluated in the same patient”

RESPONSE: We appreciate the suggestion given by the reviewer. We reviewed the text and some details of the statistical analysis. We took the opportunity to make some changes to the figures to make our results clearer for the reader. The explanation for the statistical analysis performed is detailed below, however, in the text of the manuscript we left it more succinct to make it clearer for the readers.

In order to compare ventilation parameters and Delta Z measurements between the study periods, we need to consider the dependence between the different measurements performed in the same period and between the different periods for the same child. The ventilation parameters (Table 2) show little or no variation in the same period (epochs), so to compare the mean of these parameters between the different periods we used generalized estimation equation (GEE) models. Delta Z measurements showed great variability between measurements of the same period and for this reason generalized mixed models were adjusted to make it possible to model the correlation structure of the random effect (patient) and also of the repeated factor (epochs). 

As requested by the reviewer, a reference was inserted.

Reference: Faraway JJ. Extending the linear Model with R: Generalized Linear, Mixed Effects and Nonparametric Regression Models, Boca Raton: Chapman & Hall/CRC, 2006.

TEXT: “The data were described as absolute and relative frequencies for the categorical variables and as medians and quartiles, in addition to minimum and maximum values for the numerical variables. The means of the mechanical ventilation parameters were obtained by fitting models of generalized estimation equations. Changes in global and regional ventilation were tested with linear mixed models, accounting for the repeated measurements with a random intercept. In addition, models were adjusted with subjects nested within time (induction-SB, MV-5min, MV-30min, MV-late, recovery-SB), considering the multiple measurements (breaths) from the same subject at each time point. The results of the models are presented as the mean values estimated by the models with corresponding 95% confidence intervals and p values (17). The analyses were performed with the R statistical software (R Foundation for Statistical Computing, Vienna, Austria) considering a significance level of 5%.”

MANUSCRIPT LOCATION: Section Methods, page 09. We modified figures 2 and 3 as well as their legends.

(2) Please provide explanations for greater ventilation in the posterior region during SB before anesthesia.

RESPONSE: We appreciate the reviewer's inquiry and the opportunity to respond. 

The literature has a very solid physiological foundation regarding regional differences in ventilation. During active inspiration, the dorsal part of the diaphragm has a greater range of incursion. In this region, the muscle fibers are more elongated, therefore, they can develop more strength (or shortening of the fibers) during inspiration. With greater amplitude of incursion, the variation of pleural pressure is greater, leading to greater ventilation of the posterior region. However, as soon as the effect of inhalation anesthesia and muscle relax take, ventilation is shifted toward the nondependent lung. One possible explanation is that during paralysis and in supine position, the anterior chest wall is more readily expanded as the posterior chest wall is splinted by the surface the patient is lying on and the cranial movement of the anterior diaphragm. During anesthesia of adult patients, the shape of the chest changes with a reduction in the cross-sectional area of the ribcage.

We have inserted a new reference in the manuscript and a new paragraph in the discussion to make the explanation more consistent.

TEXT: “During spontaneous breathing, diaphragmatic contraction has a strong influence on the direction of ventilation. The posterior portion of the dome presents greater amplitude during the incursion and, therefore, greater pressure variation in the posterior region of the thorax. During active inspiration, the dorsal part of the diaphragm has a greater range of incursion. In this region, the muscle fibers are more elongated, therefore, they can develop more strength during inspiration. With greater amplitude of incursion, the variation of pleural pressure is greater, leading to greater ventilation of the posterior region. However, as soon as the muscle relax take, ventilation is shifted toward anterior region. That because is during muscle relax and in supine position, the anterior chest wall is more readily expanded as the posterior chest wall is splinted by the surface the patient is lying on and the cranial movement of the anterior diaphragm. During anesthesia of adult patients, the shape of the chest changes with a reduction in the cross-sectional area of the ribcage.”

REFERENCES

Froese AB. Anesthesia-paralysis and the diaphragm: in pursuit of an elusive muscle. Anesthesiology. 1989 Jun;70(6):887-90. PMID: 2729628.

Reber A, Nylund U, Hedenstierna G. Position and shape of the diaphragm: Implications for atelectasis formation. Anaesthesia. 1998;53(11):1054–61. 

Hedenstierna G, Lichtwarch-Aschoff M. Interfacing spontaneous breathing and mechanical ventilation. New insights. Minerva Anestesiol. 2006;72(4):183–98. 

Wahba RWM. Perioperative functional residual capacity. Can J Anaesth. 1991;38(3):384–400. 

MANUSCRIPT LOCATION: Section Discussion, page 13

(3) Please provide more explanations about the increased Delta Z during CMV. What was the mode of MV?

We thank the reviewer for the question. The ventilatory modality used in all patients was pressure controlled and it is worth mentioning that, as this was an observational study, there was no control over the ventilatory parameters used by the anesthesiologists. In our study, as described in our discussion, there was an increase in Delta Z after starting positive pressure ventilation compared to spontaneous breathing at the time of anesthetic induction. The literature corroborates our findings and shows that, compared to spontaneous breathing, the use of positive inspiratory pressures increased Delta Z by up to 50%. 

Radke OC, Schneider T, Vogel E, Koch T. Effect of trigger sensitivity on redistribution of ventilation during pressure support ventilation detected by electrical impedance tomography. Anesthesiol Pain Med. 2015 Aug 1;5(4). 

Humphreys S, Pham TMT, Stocker C, Schibler A. The effect of induction of anesthesia and intubation on end-expiratory lung level and regional ventilation distribution in cardiac children. Paediatr Anaesth. 2011 Aug;21(8):887–93.

TEXT: “The analysis of the impedance variation, Delta Z, showed an increase in whole ventilation with the beginning of mechanical ventilation, a fact previously described once the use of positive inspiratory pressure results in higher tidal volumes compared to spontaneous breathing (27-28). Humphreys et al evaluating children undergoing cardiac surgery showed that the delta z almost doubles with the start of mechanical ventilation when compared to spontaneous breathing (28).”

MANUSCRIPT LOCATION: Section Discussion, page 15

(4) Please provide more explanations about the still-increased Delta Z during SB after surgery.

During the period of controlled mechanical ventilation, 95% of our patients received opioids (fentanyl). Radke, in his study that compared the distribution of ventilation in during spontaneous breathing and in throughout mechanical ventilation, points out that the use of opioids for reducing the respiratory rate (RR)can influence the tidal volume (Vt). So, for maintaining a similar minute volume, a diminished RR would determine a higher Vt. 

We speculate that the residual fentanyl effect during recovery time might relate to Delta Z increment in comparison to the induction time. However, no statistical significance was observed between the induction-SB and recovery-SB epochs. 

Journal Requirements:

RESPONSE: All text has been proofread and formatted according to the PlosOne’s style.

2. You indicated that you had ethical approval for your study. In your Methods section, please ensure you have also stated whether you obtained consent from parents or guardians of the minors included in the study or whether the research ethics committee or IRB specifically waived the need for their consent

RESPONSE: We acknowledge for this and we have inserted in the “Methods section” the statement “An informed consent form was obtained from each child’s parents.”

RESPONSE: We recognize this would be appropriate but unfortunately in Brazil we have a big restriction on data availability due to the data protection law and our institution does not have any registration with specific platforms for data deposit.

RESPONSE: We reinforce that no funding or scholarship was granted for this study, but we will review the data in the ‘Funding Information’ section

---

## [Editor Report · Decision Letter 1]

1 Nov 2022

PONE-D-22-15065R1Effect of anesthesia and controlled mechanical ventilation on pulmonary ventilation distribution assessed by EIT in healthy childrenPLOS ONE

Dear Dr. Nascimento,

Thank you for submitting your manuscript to PLOS ONE. After careful consideration, we feel that it has merit but does not fully meet PLOS ONE’s publication criteria as it currently stands. Therefore, we invite you to submit a revised version of the manuscript that addresses the points raised during the review process.

What is "EIT" on your title?  Please never use acronyms or abbreviations upfront on your Title!

We look forward to receiving your revised manuscript.

Kind regards,

Academic Editor

PLOS ONE

Additional Editor Comments:

Please don't use acronyms or abbreviations upfront on the Title! What is EIT? Very confusing!
---

## [Author Response · Author response to Decision Letter 1]

16 Nov 2022

Although our institution does not have any registration on specific platforms for data deposit, we are sharing our data as Supporting Information files. We appreciate the opportunity to be able to provide them in this way.

This manuscript is the result of a doctoral thesis that finishs on November 23rd. It would be very important, if possible, if we get the answer before this date.

---

## [Decision Letter · Decision Letter 2]

26 Dec 2022

PONE-D-22-15065R2Effect of anesthesia and controlled mechanical ventilation on pulmonary ventilation distribution assessed by electrical impedance tomography in healthy childrenPLOS ONE

Dear Dr. Nascimento,

Thank you for submitting your manuscript to PLOS ONE. After careful consideration, we feel that it has merit but does not fully meet PLOS ONE’s publication criteria as it currently stands. Therefore, we invite you to submit a revised version of the manuscript that addresses the points raised during the review process.

Please revise.

We look forward to receiving your revised manuscript.

Kind regards,

Academic Editor

PLOS ONE

Reviewers' comments:

Reviewer's Responses to Questions

**Comments to the Author**

1. If the authors have adequately addressed your comments raised in a previous round of review and you feel that this manuscript is now acceptable for publication, you may indicate that here to bypass the “Comments to the Author” section, enter your conflict of interest statement in the “Confidential to Editor” section, and submit your "Accept" recommendation.

Reviewer #2: All comments have been addressed

Reviewer #3: (No Response)

2. Is the manuscript technically sound, and do the data support the conclusions?

Reviewer #2: Yes

Reviewer #3: Yes

3. Has the statistical analysis been performed appropriately and rigorously? 

Reviewer #2: Yes

Reviewer #3: Yes

4. Have the authors made all data underlying the findings in their manuscript fully available?

Reviewer #2: Yes

Reviewer #3: Yes

5. Is the manuscript presented in an intelligible fashion and written in standard English?

Reviewer #2: Yes

Reviewer #3: Yes

6. Review Comments to the Author

Reviewer #2: Excellent job on your second revision , children and clinicians will benefit greatly. Happy Holidays.

Reviewer #3: In this manuscript, Nascimento et al. describe the effect of general anesthesia on pulmonary ventilation distribution in children. The study was performed by experts in the field, was well conducted and is well written. However, the reviewer has some major concerns:

- What ventilatory mode was applied during the different epochs?

- Were the patients allowed to breathe spontaneously during the different epochs?

- Please provide (static) compliance during all epochs. I suggest that you will find significant changes. If this is not the cause, the amount of atelectasis may not be that great.

- Could you please provide an EIT analysis (Figure) using the center of ventilation? Maybe the reader can then decide which description is more intuitive.

- How would you change the ventilatory settings according to EIT measurements in respect of avoiding atelectasis, alveolar cycling and alveolar overdistension?

Minor concerns:

- Table 2: Please define all abbreviations. What do you mean with irpm? Please include tidal volume / predicted body weight

- How can you assure that you included only lung-healthy patients?

- Title: Please use the term “general anaesthesia”

- I suggest using measurement periods may be more appropriate than epochs

- What does the reader really learn from Figure 2?

7. PLOS authors have the option to publish the peer review history of their article (what does this mean?). If published, this will include your full peer review and any attached files.

Reviewer #2: **Yes: **Joseph Schlesinger

Reviewer #3: No

---

## [Author Response · Author response to Decision Letter 2]

27 Jan 2023

Reviewers' Comments to Authors:

We appreciate the reviewers' time and effort dedicated to our manuscript, their comments and suggestions have certainly helped us to improve our work. Below is a point-by-point answer to all the questions raised by the reviewers.

Reviewer #3: In this manuscript, Nascimento et al. describe the effect of general anesthesia on pulmonary ventilation distribution in children. The study was performed by experts in the field, was well conducted and is well written. However, the reviewer has some major concerns:

- What ventilatory mode was applied during the different epochs?

RESPONSE: We appreciate the reviewer's question. The ventilatory mode applied during all studied periods of mechanical ventilation was pressure controlled ventilation. We inserted a phrase in the “Results” section highlighting this aspect.

TEXT: Pressure Controlled ventilatory modality was used in all patients and no patient triggered cycle was detected during the analyzed periods. 

MANUSCRIPT LOCATION: Section “Results”, page 10.

- Were the patients allowed to breathe spontaneously during the different epochs?

RESPONSE: No, they were not allowed to spontaneously ventilate during the study period. As previously mentioned, all patients remained on controlled ventilation and did not have spontaneous cycles. 

TEXT: Pressure Controlled ventilatory modality was used in all patients and no patient triggered cycle was detected during the analyzed periods. 

MANUSCRIPT LOCATION: Section “Results”, page 10.

- Please provide (static) compliance during all epochs. I suggest that you will find significant changes. If this is not the cause, the amount of atelectasis may not be that great.

RESPONSE: We appreciate the reviewer's comments and compliance was obtained, and the values were entered in Table 2. There was no significant difference in compliance results when comparing the 3 periods of mechanical ventilation.

TEXT: The delta pressure (Pins - PEEP), VT adjusted for patient weight (in mL/kg) and compliance (mL/cmH2O) were calculated.

MANUSCRIPT LOCATION: Section “Methods”, page 08 and Section “Results”, table 2.

- Could you please provide an EIT analysis (Figure) using the center of ventilation? Maybe the reader can then decide which description is more intuitive.

RESPONSE: We appreciate the reviewer's comment. The figure with the CoV values was created and designate Figure 3. We inserted a sentence in methods.

TEXT: The distribution of ventilation in percentage was presented as dorsal ventilation fraction, which was calculated by the ratio of the Delta Z of the posterior region by the global Delta Z multiplied by 100 and by center of ventilation (CoV) that is defined based on the center of gravity or center of mass and its value is given automatically by the EIT (17). (Methods) 

When ventilation distribution was evaluated by CoV the results were like those found by the dorsal ventilation fraction. There was redistribution from the posterior to the anterior region observed with the onset of the MV (posterior ventilation went from 51% (IC95%: 49-53%) to 47% (IC95%:46-49%) (p<0.001). With the restoration of spontaneous breathing, ventilation distribution returned towards the normal pattern with predominant ventilation in the dorsal regions (posterior ventilation went from 48% (IC95%: 47-50%) to 49% (IC95%:47-51%), although a small difference persisted as compared to the induction-SB period but without significance (Fig 3). (Results)

Figure 3: Distribution of pulmonary ventilation assessed by center of ventilation (CoV) presented over the five time points: induction-SB, MV-5min, MV-30min, MV-late and recovery-SB. The CoV is a parameter that quantifies the distribution of ventilation. When we have a value of 50 it means that ventilation is equally distributed between the anterior and posterior regions of the thorax. Higher numbers indicate a shift towards the dorsal region, and lower numbers indicate a shift towards the ventral region. Significant variations were observed in the ventilation distribution measures demonstrated in percentage when compared MV-5min, MV-30min and MV-late to induction-SB.

MANUSCRIPT LOCATION: Section “Methods”, page 08; Section “Results”, page 12 and figure legends.

- How would you change the ventilatory settings according to EIT measurements in respect of avoiding atelectasis, alveolar cycling and alveolar overdistension?

RESPONSE: The EIT presents several resources for monitoring and adequacy of mechanical ventilation. Among the most used EIT tools at the bedside we emphasize the ventilation distribution map (given in percentage), and the Delta Z, which has a strong correlation with tidal volume (Vt). 

Checking alveolar cycling can be done also by the color scale automatically performed by the EIT apparel, where dark blue color relates to low Vt and lighter blue to white relates to high tidal volumes. This is done in real-time by the device, considering the relative Delta Z variation at each cycle. 

To perform an EIT ventilation evaluation it is possible to elect the regions of interest (ROIs), such as anterior and posterior, in the vertical axis, for instance. However, a caution for this interpretation includes knowledge of how the behavior of a healthy lung is when exposed to mechanical ventilation and this was the main objective of this study. 

Once we know that the distribution of anterior-posterior ventilation of healthy lung children on mechanical ventilation is quite similar, about 50% of Vt directed to the anterior and posterior ROIs, we can target this distribution when we adjust the ventilatory parameters at the bedside. 

But, for instance if we observe a posterior ventilation distribution lower than 50% of the Vt and progressively decreasing, this suggests an ongoing collapse in this region, and then PEEP rise might be a good strategy. On the other hand, if we observe a posterior ventilation distribution above 50% with the child on mechanical ventilation, we can interpret that the anterior region is hyperinflated, since the non-dependent lung is more prone to earlier overdistension, and the adjustment of parameters such as PEEP reduction or tidal volume adjustment is necessary.

 In addition, observation of Delta Z during ventilation provides information on Vt and regional ventilation. We may be ventilating with the theoretical concept of protective ventilation, however, when we look at regional ventilation, there are regions that are receiving much higher volumes than others when the lung aeration is not adequately titrated. 

We consider that EIT allows an additional and different layer of monitoring, making ventilation safer and more effective.

With those concepts in mind, EIT will help tailoring ventilatory parameters at the bedside in a unique approach, likely minimizing atelectrauma and volutrauma, guiding when titrating ventilatory parameters at the bedside, monitoring patient decubitus changes and recruitment maneuvers. 

We have inserted a sentence in the discussion section to make it clearer for the reader.

TEXT: Once we know that in children with healthy lungs the anterior-posterior ventilation distribution during mechanical ventilation is quite similar, we might titrate the ventilatory parameters to maintain this distribution pattern. In accordance to this concept, if posterior zones ventilation is going below 50%, collapse might be present and then PEEP increase and FRC gain shall be the target. On the other hand, if posterior ventilation distribution is above 50%, anterior hyperinflation is expected and parameters adjustments such as PEEP reduction or tidal volume adjustment are likely necessary. 

MANUSCRIPT LOCATION: Section “Discussion”, page 15.

Minor concerns:

- Table 2: Please define all abbreviations. What do you mean with irpm? Please include tidal volume / predicted body weight

RESPONSE: All abbreviations have been inserted and the respiration rate unit has been adjusted. Tidal volume by weight was already in the table.

MANUSCRIPT LOCATION: Table 2.

- How can you assure that you included only lung-healthy patients?

RESPONSE: Children were eligible for the study presenting no clinical symptoms that would result in impact over the lungs, and only after evaluation by the anesthesiologist. In the anesthetist's evaluation, the parents were asked about a possible history of prematurity, prior or recent episodes of wheezing and about other chronic diseases such as cystic fibrosis. Thus, it was possible to admit that all included children did not present pulmonary diseases. 

- Title: Please use the term “general anaesthesia”

RESPONSE: The term “general anesthesia” has been included in the title

TEXT: Effect of general anesthesia and controlled mechanical ventilation on pulmonary ventilation distribution assessed by electrical impedance tomography in healthy children.

MANUSCRIPT LOCATION: Title

- I suggest using measurement periods may be more appropriate than epochs

RESPONSE: All text has been revised and all terms "epochs" have been replaced by "periods"

MANUSCRIPT LOCATION: All text

- What does the reader really learn from Figure 2?

RESPONSE: The Figure 2 shows the distribution of pulmonary ventilation assessed by the dorsal ventilation fraction in the anterior and posterior regions presented. It Shows that there was a redistribution of ventilation from the posterior to the anterior region with the onset of mechanical ventilation and with the restoration of spontaneous breathing, ventilation distribution returned towards the normal pattern with predominant ventilation in the dorsal regions. When we use the EIT to evaluate the distribution of pulmonary ventilation, we expect to find a more homogeneous ventilation, that is, distribution of anteroposterior ventilation close to 50% anterior and 50% posterior.

TEXT: There was a redistribution of ventilation from the posterior to the anterior region with the onset of MV (posterior ventilation went from 54% (IC95%: 49-60%) to 49% (IC95%:44-53%) (p = 0.002) and remained stable over the whole ventilation period (Fig 2). With the restoration of spontaneous breathing, ventilation distribution returned towards the normal pattern with predominant ventilation in the dorsal regions (posterior ventilation went from 51% (IC95%: 42-61%) to 56% (IC95%:51-62%) (p = 0.02), although a small difference persisted as compared to the induction-SB period but without significance (Fig 2). 

MANUSCRIPT LOCATION: Section “Results”, page 11.

---

## [Decision Letter · Decision Letter 3]

22 Feb 2023

PONE-D-22-15065R3Effect of general anesthesia and controlled mechanical ventilation on pulmonary ventilation distribution assessed by electrical impedance tomography in healthy childrenPLOS ONE

Dear Dr. Nascimento,

Thank you for submitting your manuscript to PLOS ONE. After careful consideration, we feel that it has merit but does not fully meet PLOS ONE’s publication criteria as it currently stands. Therefore, we invite you to submit a revised version of the manuscript that addresses the points raised during the review process.

Please revise.

We look forward to receiving your revised manuscript.

Kind regards,

Academic Editor

PLOS ONE

Journal Requirements:

Reviewers' comments:

Reviewer's Responses to Questions

**Comments to the Author**

1. If the authors have adequately addressed your comments raised in a previous round of review and you feel that this manuscript is now acceptable for publication, you may indicate that here to bypass the “Comments to the Author” section, enter your conflict of interest statement in the “Confidential to Editor” section, and submit your "Accept" recommendation.

Reviewer #3: All comments have been addressed

Reviewer #4: All comments have been addressed

2. Is the manuscript technically sound, and do the data support the conclusions?

Reviewer #3: Yes

Reviewer #4: Yes

3. Has the statistical analysis been performed appropriately and rigorously? 

Reviewer #3: Yes

Reviewer #4: I Don't Know

4. Have the authors made all data underlying the findings in their manuscript fully available?

Reviewer #3: Yes

Reviewer #4: Yes

5. Is the manuscript presented in an intelligible fashion and written in standard English?

Reviewer #3: Yes

Reviewer #4: Yes

6. Review Comments to the Author

Reviewer #3: Thank you very much for your revision. You did a great job and I have no further comments!

Best regards!

Reviewer #4: In this manuscript, effects of general anesthesia on pulmonary ventilation distribution in children are examined . The study was performed by EIT. The study was by experts in the field, was well conducted and is well written. The previous comments and changes appear to have improved the quality, too. Some comments:

1. The study has been performed in 20 patients. Although this has been stated as a limitation, I wonder whether the study is "powered" enough to make some conclusion. My knowledge in statistics is not good enough to understand whether and how a comparison of 5 measurement times was possible in distribution of ventilation.

2. similarly, I would wonder whether different ventilation strategies would lead to different distributions of ventilations; but first of all, the strategy is more or less very similar (i.e. almost the same PEEP) , and second, sample size is too low for any assumption.. But at least, has there been any recruitment manover in any of the patients in any time?

3. Difference adult- child: there is some classical info written in the discussion.. But i wonder : what are the differences in EIT between adult and child? can you discuss the previous study made in adults with this one in child? (I am aware that a "comparison" is not possible, but the reader would like to know a simple thing: is the change caudes by mechanical ventilaton less or more in children? at least, maybe some speculations)

7. PLOS authors have the option to publish the peer review history of their article (what does this mean?). If published, this will include your full peer review and any attached files.

Reviewer #3: No

Reviewer #4: **Yes: **Mert Senturk

---

## [Author Response · Author response to Decision Letter 3]

23 Feb 2023

Reviewers' Comments to Authors:

We appreciate the reviewers' time and effort dedicated to our manuscript, their comments and suggestions have certainly helped us to improve our work. Below is a point-by-point answer to all the questions raised by the reviewers.

Reviewer #4: In this manuscript, effects of general anesthesia on pulmonary ventilation distribution in children are examined. The study was performed by EIT. The study was by experts in the field, was well conducted and is well written. The previous comments and changes appear to have improved the quality, too. Some comments:

1. The study has been performed in 20 patients. Although this has been stated as a limitation, I wonder whether the study is "powered" enough to make some conclusion. My knowledge in statistics is not good enough to understand whether and how a comparison of 5 measurement times was possible in distribution of ventilation.

RESPONSE: We appreciate the reviewer's comments and this question has been shared with our team statistician. This issue was a concern of the authors when the study was idealized. However, this was the first study that would address the effect of anesthesia on ventilation distribution in the pediatric population. As there is no reference to pediatric values of the EIT data evaluated for sample calculation, it was decided to carry out a convenience sample. The problem in cases of small samples is that a type 2 or beta error may occur, which shows no significant difference when in fact there is such a difference. However, our results pointed to a significant difference in the distribution of ventilation (which was our main analysis variable), thus pointing to a low probability of type 2 error. We included a sentence at the end of the statistical analysis to make it clear for the reader. 

Regarding the 5 evaluation moments, all moments were compared with the Induction-SB moment through linear mixed models. Having a large number of evaluations in a small sample is not really a problem. in addition, we considering the multiple measurements (breaths) from the same subject at each time pointe, this increases the power of the analysis as it enables a better fit of the model.

TEXT As there is no reference to pediatric values of the EIT data evaluated for sample calculation, it was decided to carry out a convenience sample.

Changes in global, regional ventilation and CoV were tested with linear mixed models, accounting for the repeated measurements with a random intercept. In addition, models were adjusted with subjects nested within time (induction-SB, MV-5min, MV-30min, MV-late, recovery-SB), considering the multiple measurements (breaths) from the same subject at each time point. The results of the models are presented as the mean values estimated by the models with corresponding 95% confidence intervals and p values (18).

MANUSCRIPT LOCATION: Section “Materials and methods”, page 9.

2. similarly, I would wonder whether different ventilation strategies would lead to different distributions of ventilations; but first of all, the strategy is more or less very similar (i.e. almost the same PEEP), and second, sample size is too low for any assumption.. But at least, has there been any recruitment manover in any of the patients in any time?

RESPONSE: We appreciate the reviewer's comment. The use of spontaneous breathing during mechanical ventilation has been discussed, in adults, as a strategy to prevent redistribution of ventilation in dependent zones, however, with controversial results. But the objective of our study was not to compare modalities and therefore, all our patients were ventilated in pressure-controlled mode without spontaneous cycles during mechanical ventilation. Despite our small sample size, our results show a significant difference in the distribution of ventilation with the use of controlled mechanical ventilation when compared with spontaneous breathing (without mechanical ventiltion) and this supports our conclusion. Inserted a sentence and 2 references in the discussion to complement the ventilatory strategy.

TEXT The use of spontaneous breathing during mechanical ventilation has been discussed as a strategy to prevent the redistribution of ventilation in dependent zones, however, with controversial results (20,21).

MANUSCRIPT LOCATION: Section “Discussion”, page 13 and References.

20. Radke OC, Schneider T, Heller AR, Koch T. Spontaneous breathing during general anesthesia prevents the ventral redistribution of ventilation as detected by electrical impedance tomography: A randomized trial. Anesthesiology. 2012;116(6):1227–34.

21. Putensen, Christian; Muders, Thomas; Varelmann, Dirk; Wrigge H. The impact of spontaneous breathing during mechanical ventilation. Curr Opin Crit Care. 2006;12(1):13–8

3. Difference adult- child: there is some classical info written in the discussion.. But i wonder : what are the differences in EIT between adult and child? can you discuss the previous study made in adults with this one in child? (I am aware that a "comparison" is not possible, but the reader would like to know a simple thing: is the change caudes by mechanical ventilaton less or more in children? at least, maybe some speculations)

RESPONSE: We appreciate the reviewer's questions. The central question of our study was whether there would be differences in EIT between adult and child. Adults present redistribution of ventilation from the posterior region to the anterior region with the onset of anesthesia and controlled mechanical ventilation. Our results found that the child has the same pattern of redistribution.

Therefore, in the redistribution of ventilation, there are no differences between adults and children, both present a more homogeneous ventilation (around 50%) with the onset of mechanical ventilation.

Considering that the posterior distribution in adults is around 60% and in children 54%, perhaps the magnitude in percentage of redistribution would be greater in adults, but only speculation.

TEXT There was a redistribution of ventilation from the posterior to the anterior region with the onset of MV (posterior ventilation went from 54% (IC95%: 49-60%) to 49% (IC95%:44-53%) (p = 0.002) and remained stable over the whole ventilation period.(Results)

All text discussion

MANUSCRIPT LOCATION: Section “Results”, page 11 and “Discussion”

---

## [Decision Letter · Decision Letter 4]

1 Mar 2023

Effect of general anesthesia and controlled mechanical ventilation on pulmonary ventilation distribution assessed by electrical impedance tomography in healthy children

PONE-D-22-15065R4

Dear Dr. Nascimento,

We’re pleased to inform you that your manuscript has been judged scientifically suitable for publication and will be formally accepted for publication once it meets all outstanding technical requirements.

Kind regards,

Academic Editor

PLOS ONE

Additional Editor Comments (optional):

Reviewers' comments:

Reviewer's Responses to Questions

**Comments to the Author**

1. If the authors have adequately addressed your comments raised in a previous round of review and you feel that this manuscript is now acceptable for publication, you may indicate that here to bypass the “Comments to the Author” section, enter your conflict of interest statement in the “Confidential to Editor” section, and submit your "Accept" recommendation.

Reviewer #3: All comments have been addressed

Reviewer #4: All comments have been addressed

2. Is the manuscript technically sound, and do the data support the conclusions?

Reviewer #3: Yes

Reviewer #4: Yes

3. Has the statistical analysis been performed appropriately and rigorously? 

Reviewer #3: Yes

Reviewer #4: Yes

4. Have the authors made all data underlying the findings in their manuscript fully available?

Reviewer #3: Yes

Reviewer #4: Yes

5. Is the manuscript presented in an intelligible fashion and written in standard English?

Reviewer #3: Yes

Reviewer #4: Yes

6. Review Comments to the Author

Reviewer #3: (No Response)

Reviewer #4: These were some minor comments for a well-prepared well-written study. Thank you for the revised version.

7. PLOS authors have the option to publish the peer review history of their article (what does this mean?). If published, this will include your full peer review and any attached files.

Reviewer #3: No

Reviewer #4: No

---

## [Editor Report · Acceptance letter]

7 Mar 2023

PONE-D-22-15065R4 

Effect of general anesthesia and controlled mechanical ventilation on pulmonary ventilation distribution assessed by electrical impedance tomography in healthy children 

Dear Dr. Nascimento:

I'm pleased to inform you that your manuscript has been deemed suitable for publication in PLOS ONE. Congratulations! Your manuscript is now with our production department. 

Kind regards, 

on behalf of

Dr. Robert Jeenchen Chen 

Academic Editor

PLOS ONE